# A Review on Multiscale Bone Damage: From the Clinical to the Research Perspective

**DOI:** 10.3390/ma14051240

**Published:** 2021-03-05

**Authors:** Federica Buccino, Chiara Colombo, Laura Maria Vergani

**Affiliations:** Department of Mechanical Engineering (DMEC), Politecnico di Milano, Via La Masa 1, 20154 Milano, Italy; federica.buccino@polimi.it (F.B.); chiara.colombo@polimi.it (C.C.)

**Keywords:** age-related bone fractures, multiscale imaging, bone damage, computational models, experimental validation

## Abstract

The investigation of bone damage processes is a crucial point to understand the mechanisms of age-related bone fractures. In order to reduce their impact, early diagnosis is key. The intricate architecture of bone and the complexity of multiscale damage processes make fracture prediction an ambitious goal. This review, supported by a detailed analysis of bone damage physical principles, aims at presenting a critical overview of how multiscale imaging techniques could be used to implement reliable and validated numerical tools for the study and prediction of bone fractures. While macro- and meso-scale imaging find applications in clinical practice, micro- and nano-scale imaging are commonly used only for research purposes, with the objective to extract fragility indexes. Those images are used as a source for multiscale computational damage models. As an example, micro-computed tomography (micro-CT) images in combination with micro-finite element models could shed some light on the comprehension of the interaction between micro-cracks and micro-scale bone features. As future insights, the actual state of technology suggests that these models could be a potential substitute for invasive clinical practice for the prediction of age-related bone fractures. However, the translation to clinical practice requires experimental validation, which is still in progress.

## 1. Introduction

Age-related bone fractures represent an increasing health and economic concern. Projections of age-related fractures in the United States show an increase of 50% in twenty years, starting from a value of 2.1 million in 2005 [1]. This evident increment is strongly related to the increase in the average age of population. Indeed, with the aging process, human bone becomes more brittle and more prone to fracture [2,3]. Age-related bone fractures can be mainly caused either by a traumatic accident or by a pathology. As regards bone pathologies, the most widespread one is osteoporosis, which induces a substantial reduction in bone strength and a higher susceptibility to fracture [4]. The most fracture-affected sites are thoracic and lumbar vertebrae [5]. In particular, with the aging process, there is an evident increase in the incidence of vertebral fractures, especially in the female population (Figure 1).

The number of women exceeding a fracture threshold is six times higher than for men. This parameter, i.e., the fracture threshold, is defined as the 10 years probability of a major fracture equivalent to that of a woman with no other clinical risk factors [6]. This gender difference could also be explained by introducing the role played by osteoporosis, that is four times more common in females than in males [7]. This is due to their smaller bone size and the advent of menopause, since osteoporosis is closely related to a lack of estrogen [8]. Often, when fractures extend to the vertebral bodies or to the hip, patients cannot be treated in a day hospital, but need to be bedridden, involving medical and nursing staff for assistance. This raises not only an enormous health concern, but also an economic burden for disease management. The health problem is also connected with the concept of frailty, defined as an increased vulnerability due to long-term hospitalization for fracture treatment, impaired quality of life and disability [9]. As regards economic aspects, a projection of worldwide direct and indirect expenses related to the identification and treatment of hip osteoporotic fractures is expected to rise almost to USD 100 billion in sixty years, starting from an evaluation of USD 34.8 billion in 1990 [4]. An overview of the economic impact of fracture treatment and fracture prevention in different countries [10] is reported in Figure 2 [11].

In order to resize the impact of age-related bone fractures and to start reducing their health and economic influence, early diagnosis is key. However, it is particularly difficult to predict bone fractures and to define different levels for bone fragility. In fact, there are several parameters that play a role when considering bone fractures. First, bone is characterized by a complex hierarchical structure [12]. The macro-scale is the whole bone level, while the meso-scale is represented by cortical and trabecular structures. Osteons are the structural units of the compact bone, whereas the trabeculae form a porous network in the spongy bone. Both cortical and trabecular bone are constituted by micrometric lamellae, composed of preferentially oriented collagen fibrils. Inside the fibrils there are lacunae, sub-micrometric cavities where osteocytes reside. At the nano-scale, collagen fibrils mainly consist of collagen type I and hydroxyapatite crystals. This intricate architecture is reflected in fracture patterns; indeed, damage occurs at different scales (Figure 3) [13]. 

In order to deal with this complexity, Ritchie et al. [14] suggest considering bone as an engineering material, subjected to different loading modes. In fact, in vivo, bones experience complex multiaxial loadings [15], comprising contributions from tension, compression and shear. However, tensile loading conditions represent the worst-case scenario for most materials [16]. Another important aspect is related to the number of loading applications; there is, in fact, an increasing relevance of fractures that happen not only due to a single overload, but also in the presence of cyclic loading [17] and in association with a deterioration of bone architecture [18]. With age, the imbalance between the resorption and deposition of matrix by osteoblasts and osteoclasts leads to cortical thinning and to increases in cortical porosity [19]. By reducing the thickness of both cortical and trabecular bone, the skeleton decreases its ability to sustain loads. This results in an increased difficulty in withstanding cyclic daily loads, typical of habitual activities such as walking [20].

It is therefore crucial to deeply understand the mechanisms of bone damage leading to bone failure. Although several studies have been performed to evaluate fracture characteristics [21], a uniquely accepted bone failure criterion is still missing. Bone shows the least resistance to fracture under mode I, purely tensile, with respect to mode II (shear loading) and mode III (tearing or anti-plane shear loading) [18]. The application of fracture mechanics concepts is an interesting way to analyze these fracture modalities, interpreting bone fracture as the propagation of cracks. In this context, it is useful to firstly comprehend the evolution of cracks from nucleation to damage accumulation at different scales. In particular, the difficulty in understanding the mechanisms of bone damage, especially at smaller scales [22], lies in the identification of the contribution of micro- and macro-scale features to crack initiation, propagation and fracture. It has been demonstrated that bone fracture is locally strain controlled [23]. An additional degree of complexity derives from bone’s ability to remodel itself: this process of synthesis (ossification) and destruction (resorption) is governed by osteoblasts and osteoclasts, respectively. Bone remodeling [24] is a process that occurs over time in the living tissue, so it is often neglected when in vitro studies are performed. This implies that bone fractures heal depending on their severity. However, for the purpose of this review, the process of bone remodeling is a secondary aspect that will not be deepened.

At all bone hierarchical scales, bone toughness deteriorates dramatically with aging [25] and it is necessary to identify proper imaging techniques to visualize changes in bone morphology. This is essential in order to measure and predict bone fragility. Several imaging techniques at the multiscale [26] are currently available: in general, clinical images can be collected with radiography, dual X-ray absorptiometry (DXA) or magnetic resonance imaging (MRI). These techniques offer an overview of the whole bone with a lower resolution, compared, for example, to micro-computed tomography (micro-CT) [27]. This latter technique increases the resolution but reduces the analyzed region. The different contribution of imaging techniques in identifying bone fractures at the multiscale (fracture at the macro-scale, micro-cracks at the micro-scale) is the first point that will be discussed in the current review. For example, a common clinical outcome from DXA is the evaluation of bone mineral density (BMD), which is an indicator of bone quantity: low BMD values indicate the presence of osteopenia or osteoporosis, so a bone that is more prone to fracture. However, it has been demonstrated [28,29] that, in the case of vertebral fractures, BMD can only predict the 70% of them [30]. BMD is therefore a limited predictor of fracture risk [31,32]. It should be combined with measures of bone quality, that cover geometrical and material features of the macro- and micro-architecture [33]. For the identification of micro-architectural features and the comprehension of their role in damage propagation, micro-scale imaging techniques such as micro-CT or synchrotron radiation [34] are used in combination with subject-specific computational models [35,36]. In recent decades, they have assumed a growing importance, according to the 3R principle that proposes the replacement, reduction and refinement of animal experiments in favor of in silico models. Micro-scale observations permit us to quantify micro-architectural parameters, such as bone volume fraction, bone surface to tissue volume, degree of anisotropy, etc. [37], but also parameters related to the lacunar network [36]. The extrapolation of these micro-scale parameters could be helpful in the future to make clinical decisions about specific therapies for bone fragility treatment. A first attempt in organizing these multi-scale parameters will be presented in this review. 

The ability to quantify and predict bone fractures by means of the implemented computational damage models [38], however, needs to be assessed by means of experimental tests combined with scans of bone samples. 

The current review therefore aims firstly at comparing multiscale imaging techniques for the assessment of bone fracture, in order to clarify which are their main outcomes, potentialities and limitations. Then, the bone damage physical principle is deepened: its comprehension is crucial for the implementation of computational damage models. Using finite element (FE) models generated directly from micro-CT or synchrotron images, it is possible to perform a “virtual experiment”, able to simulate mechanical loading of the sample and to potentially predict damage formation. The last section of this review is devoted to the comparison between different validation strategies for the numerical models. This is essential to assess age-related fractures and improve fracture prediction in diseased subjects.

## 2. Imaging Techniques for Multiscale Damage Assessment and Prediction

The use of imaging techniques enables researchers to understand bone damage at different hierarchical scales. It is particularly relevant in the comprehension of the implications of fracture processes in the deterioration of bone quality. This section presents an overview of the available imaging techniques (Figure 4) to visualize bone morphologies, to assess bone fractures and to predict the fracture risk from the macro-scale to the nano-scale. The macro-architecture is currently evaluated by means of common clinical images. At lower hierarchical levels, the identification of damage processes is more complex and requires higher-resolution techniques.

Different techniques are compared in terms of outcomes, in vitro or in vivo applications, resolution, two- or three-dimensional features and the main advantages and disadvantages.

### 2.1. Macro- and Meso-Scale Imaging

Table 1 shows the principal macro-scale techniques for the imaging of bone fractures and for fracture risk prediction.

### 2.2. Micro- and Nano-Scale Imaging

Table 2 shows the principal micro- and nano-scale techniques for the imaging of bone fractures and for fracture risk prediction.

Macro-scale in vivo images are commonly used in current clinical practice; radiologic imaging is crucial in the diagnosis of fractures, because it can detect damage or abnormalities that are not identified during physical examinations. However, this technique is insensitive to even significant changes in BMD (up to 40%, as reported in Table 1). This need for higher accuracy in bone fracture diagnosis is not completely satisfied by DXA (Table 1), which is not able to catch bone features, such as thin trabeculae, where damage presumably starts. The requirement for a higher resolution for bone damage detection is in contrast with the higher radiation dose, which should be provided when increasing the detail in bone imaging. In current clinical practice, quantitative computed tomography (QCT) is used only for well-known dangerous sites such as the hip or the lumbar spine in which the possible harm from a relevant radiation dose is far lower than the expected risk from the untreated fracture. 

In recent decades, in order to find possible solutions to overcome this issue, micro- and nano-scale imaging techniques have been deeply studied with the aim to offer new tools for fracture prediction. It is interesting to point out that these in vitro techniques offer the possibility to explore micro-scale bone features and to comprehend their role in bone damage. An example is stereomicroscopy based on histological sections, that provides the possibility to assess bone damage and remodeling at a resolution higher than 2 µm, or scanning electron microscopy. However, these techniques are destructive and, in the case of stereomicroscopy, are restricted to a bidimensional analysis. The three-dimensional evaluation of bone damage at a higher resolution is crucial not only from a research perspective, but especially from the clinical point of view; the comprehension of the micro-damage physical principle and its visualization is a relevant opportunity to understand changes in the diseased tissue and to provide insights into the prevention of age-related fractures. This could be performed by the implementation of micro- and nano-scale fragility indexes, obtained with the aid of numerical models that use, as a source, micro-scale images of bone. In this context, a wide interest is devoted to micro-CT scans and synchrotron radiation (SR) images. They provide, in fact, a three-dimensional reconstruction of bone micro-architecture, which is important for the identification of microdamage, offering the optimal balance between resolution and field of view. In addition to this, the SR technique is a promising solution for the real-time visualization of bone damage, allowing the performance of mechanical tests inside the synchrotron facility. The disruptive advent of these high-resolution in vitro techniques offers the possibility to experimentally validate numerical damage models, as deeply discussed in the section “Validation approaches to multiscale damage models”. Additionally, these techniques could help the study of effective pharmacological treatments for bone pathologies. Current treatments, in fact, are administered just after an evident diagnosis of osteoporosis, often when patients have already undergone a severe fracture. Consequently, the possibility to correlate micro-scale fragility indexes with the current bone meso- and macro-architectural knowledge may support the fracture risk assessment strategies and may better address specific therapies.

## 3. Bone Damage Physical Principle

Imaging techniques are a valuable tool for the visualization of bone features and multiscale damage and could help in the comprehension of bone damage mechanisms. Due to the fact that bone is a hierarchical material with multiple linked scales, bone fracture mechanisms also occur at the multiscale. Those mechanisms are particularly complex and not completely understood. 

At the macro-scale, there are essentially two main bone fracture modalities: impact fracture and stress fracture. The former is a fracture caused by an overload, while the latter is associated with cyclic loading and it causes material failure by excessive damage accumulation [20,65]. Stress fractures include both fatigue fractures, which result from strong physical activity, and fragility fractures, which are consequences of everyday common activities. The latter are often linked to age-related decreases in bone’s ability to self-repair. This often happens in age-related bone pathologies such as osteoporosis. At the macro-scale, the shape, size and density of bone strongly affect fracture behavior [64].

At the meso-scale, trabecular bone microarchitecture influences the damage. Trabecular bone is subjected to a variety of loads during activities of daily life and the orientation of the applied load with respect to the trabecular distribution plays an important role in a possible trabecular tissue degeneration. Indeed, trabeculae generate a three-dimensional, open porous space. Pathologies such as osteoporosis may lead to a conversion from a plate-like to a rod-like trabecular morphology, which contributes to increasing the fracture risk [66]. Typically, trabecular failure initiates at the weakest trabecula or at the weakest trabecular region. From there on, the failure will progress and failure bands will develop. During a tensile test, there will be only one failure band; multiple failure bands, however, occur in compressive tests [67]. Gibson [68] investigated how the basic failure modes (bending, compression, tension, shear) can explain trabecular bone failure (Figure 5). It has been also observed that complete fractures are present in trabeculae that are oriented transversally to the direction of the applied load [69].

At the micro-scale, a complete comprehension of the bone damage physical principle is still lacking. Bone is able to sustain many micro-cracks, if they are not critical; this suggests that bone can be classified as a “damage-tolerant material” [70]. This concept is particularly interesting when applied to bone microstructure. At the micro-scale, the damage assumes two frequent morphologies: linear and diffuse damage. The linear damage appears as a sharp line of 100–200 µm [71] separating the bone matrix, and it is typical of regions subjected to compression [20]. When external compression boundary conditions are applied, linear cracks propagate parallel to the compression loading axis [72]. On the contrary, diffuse damage is characterized by a cloud of tiny cracks [73], whose dimensions are less than 1 µm [71], and this is typical of regions subjected to tension. Diffuse micro-cracks are distributed transversally to the applied tensile stress [74]. Voide et al [75] suggest that diffuse damage could be considered as linear damage when oriented differently with respect to a global reference system. Micro-cracks occur on multiple planes when subjected to torsion or mixed-mode loading (Figure 6).

Micro-cracks usually initiate at regions characterized by high stresses. As soon as the crack initiates, its growth behavior is influenced by micro-scale heterogeneities. In this context, particular interest has recently been devoted to cellular-level porosity represented by the lacunar network, whose role is still under debate. A variation in the lacunar distribution or in the lacunar shape, as happens with aging, significantly affects the bone resistance to fracture [76]. Lacunae play a dual mechanical role, having an effect on both strength and toughness. In the first instance, lacunae are stress concentrators [77], by representing bone discontinuities able to locally amplify stresses and strains. The average strain around lacunae is 1.5–4.5 times higher than the remote strain applied to the surrounding tissue [78,79]. In this sense, lacunar system should contribute to strength decreases.

However, considering bone as a damage-tolerant material, experimental investigations [75] show that lacunae are not the starting point for the micro-cracks that ultimately lead to bone failure. Micro-cracks nucleate generally at canals [75] or at cement lines and inter-lamellar zones [80,81], where the stress amplification is greater than at the lacunae [82,83]. According to those investigations, lacunae make a beneficial contribution to toughness [72]. Voide et al. suggest that lacunae exert an attraction upon the existing micro-crack: the deviation of the crack path reduces the energy of its progression, slowing down crack propagation [84]. The two effects of the lacunar network are schematized in Figure 7. The role of the lacunar network still needs to be clarified by an effective experimental validation of the proposed hypotheses, as mentioned in Section 5.

Finally, at the nano-scale, bone failure and fracture are influenced by bone’s composition in terms of hydroxyapatite and collagen fibers. Hydroxyapatite minerals show a brittle behavior and are more resistant to compression, while collagen fibers are more resistant to tension. When the crack propagates, the hydroxyapatite will fail first and the complete failure only occurs when collagen fibers are fully stretched [85].

It is understood how the comprehension of the bone damage physical principle at the multiscale level is a crucial point for the implementation of computational damage models. Those models have recently attracted a high degree of interest [86], due to the fact that patient-specific simulations, in particular in this context of age-related bone fractures, allow a more accurate prognosis [87,88]. The ability to quantify the effects of aging and pathologies such as osteoporosis is a challenging issue that computational damage models should consider.

## 4. Multiscale Computational Damage Models

Given that macro- and micro-architecture play an important role in bone damage and crack propagation, computational analyses can be interesting tools to investigate the effect of their variation due to aging on bone fractures. Using finite element (FE) models, generated directly from micro-CT or synchrotron images, it is possible to perform a “virtual experiment”, able to simulate mechanical loading of the sample and to evaluate damage formation. The main advantage of FE models is that, once the geometry of the structure is implemented, the same model can be used for multiple analyses, i.e., applying different loading conditions that may mimic everyday life loadings (e.g., walking, running).

Furthermore, bone FE models provide new insight into the relationship between damage propagation and macro- and micro-scale architecture, by allowing the localization of zones that are more prone to fracture. While in the past FE models were limited by input image resolution and computational power [87], nowadays, accurate simulations are feasible, thanks to the recent developments in imaging and in optimized computational software. The increased accuracy of bone FE damage models demonstrates that computational simulations are a reliable way to improve the choice of more effective experimental tests, potentially leading to a reduction in the number of performed tests [89,90].

In this section, both macro- and micro-scale models are presented, with an attempt to investigate bone fracture initiation and propagation at different scales [91].

Macro-scale organ-level models primarily focus on the effects of mechanical stimuli on bone resistance to fracture. Those models, that start mainly from CT reconstructions, try to implement the observation of J. Wolff [92], related to bone’s capability to adapt to the external mechanical loading conditions [93]. At this scale, micro-architectural features are neglected. The outer shape object of the problem is simplified, and filled with different meshing strategies, for instance, geometry-based with tetrahedral elements, or voxel-based with hexahedral elements. While the first method, that requires a smaller number of degrees of freedom, is not fully automated, the voxel-based hexahedron meshing has no geometrical limitations and it is a fast and completely automated mesh generation technique [94]. However, in the second case, outer surfaces or sharp geometrical discontinuities may generate a lack of convergence. Generally, the voxel-based meshing strategy allows a simpler and more effective interface with CT scans. The bone material is typically considered as a continuum solid [95]. The model is loaded with defined boundary conditions (compression, torsion or mixed-mode loading). The external load produces local stresses and strains in each considered element of the mesh (Figure 8), which could be correlated by means of the constitutive equation [96]. The constitutive equation, in the condition of linear elasticity, provides a linear relation between stresses and strains by means of the mechanical properties of the bone tissue, assuming the strains to be small or infinitesimal [97].

The initiation and propagation of damage in the macro-scale bone model is defined according to a proper failure criterion [98]. Based on the theory of elasticity, those models generally assume that bone resorption occurs when local mechanical stress overcomes a homeostatic stress state [99]. In the definition of failure at the macro-scale, it should be considered that local tissue failure could provide a loss in the structural integrity. The main results of macro-scale models, as reported in the scheme of Figure 9, show that subject-specific FE models predict values of strain with an accuracy of 90%, obtained by comparison with experimental measurements on cadaver bones (the average errors on surface strains are lower than 10%) [100]. The encouraging results are due to the coexistence of a precise geometric reconstruction, an appropriate choice of the density–elasticity relationship, an accurate application of boundary conditions and a suitable algorithm for material property evaluation. 

A relevant aspect of macro-scale models is that they can consider an accurate map of the tissue elastic modulus distribution, based on the density distribution over the continuum bone model, as performed by Viceconti [98] and Taddei [101]. These models show low average error in predicting failure load in vitro and have a high precision in identifying the location of failure [98].

Further attempts in macro-scale FE modes have been performed in order to capture the non-linear behavior of bone that depends both on the anatomic site and the loading mode. In particular, Imai et al. [102] observe that the prediction of vertebral fracture is intricate due to the complex geometry of the vertebra and its elastoplasticity. They implement non-linear FE models of the whole vertebra; in order to consider bone heterogeneity, the mechanical properties of each element are computed from the Hounsfield unit value. The correlation between the measured value of fracture strength and the predicted values with the non-linear FE models shows significant improvements [102] with respect to previous linear and simplified FE models. Other attempts in modeling bone failure have been performed by Harrison et al. [103], considering that the tissue failure consists of two phases: damage and fracture. This study develops a computational model consisting of an explicit representation of complete failure, incorporating non-linear damage criteria, fracture criteria, cohesive forces, asymmetry and large deformation capabilities.

However, those models are not able to capture patient-specific cortical nor trabecular micro-architecture, which play a crucial role in bone strength [103,104,105].

Micro-scale models provide further insights into bone fracture prediction at smaller scales. Traditionally, micro-architectural features were assessed by means of histological sections [106] that lack in three-dimensionality. Nowadays, the input of micro-scale models often comes from micro-CT or SR images, that have a resolution able to non-destructively capture not only trabecular architecture, but even ultrastructural porosities, such as lacunae. However, this corresponds to higher computational costs, calculated by van Rietbergen et al. [107] through an element-by-element method that implies the use of uniformly shaped elements to reduce memory allocation and optimize computational times. Some issues arise when dealing with micro-CT or SR imaging techniques: filtration and segmentation. Reconstructed image data include noise that should be removed or at least reduced by filtering. The choice of an adequate filter (the Gaussian filter is the most suitable one, which is easily implementable and fast in computation [108]) is essential in order to obtain an input image for the model, as close as possible to the original sample. Another relevant aspect is the correct segmentation process that selects those voxels that are below or above a defined threshold, so as to separate those elements that are bone and those that can be considered as voids. This distinction is particularly relevant when dealing with the identification of micro-cracks and lacunae, whose typical size is significantly smaller (about 10 µm [72]) with respect to the trabecular architecture. Micro-FE models are often used for the prediction of bone fracture. For this purpose, specific damage criteria are implemented. In the work of Pistoia et al. [109], a micro-FE linear analysis is performed on the human distal radius. In order to predict bone failure, the chosen criterion is that the bone failure initiation occurs when bone is strained beyond a critical value, defined as the yield strain. This model is able to give a better prediction of bone failure loads (R^2^ = 0.75 correlation with experimental testing) with respect to macro-scale investigation techniques such as DXA measurements. Linear micro-FE models are also proposed for the study of the interaction between micro-cracks and micro-porosities, such as lacunae or surface discontinuities in bones. The role of the lacunar network in damage initiation and propagation is still unclear, due to the difficulty in obtaining dynamic visualization of the advancing crack. Micro-FE models shed some light on micro damage initiation and propagation. In this context, it is necessary to define a proper descriptor of the stress distribution, such as the strain energy density (SED) or the maximum principal stress (σ_1_) criteria. SED is a good predictor for the mechanical environment sensed by the osteocytes that reside in lacunae [110] and σ_1_ is a mechanical quantity often used to assess the failure in brittle materials [111]. Those two criteria show different sites of crack initiation and different directions for crack propagation (Figure 10). 

The σ_1_ criterion locates peak stresses and strains at regions matching with the experimental initiation stresses and strains found by Voide et al. [75]. Donaldson et al. [111] compare different linear micro-FE models that aim at the identification of crack initiation sites and of the role of the lacunar network. They present a relevant novelty by implementing a stress gradient model, schematized in Figure 11, that shows a clear directionality in the advancing crack (instead of the less realistic stress limit algorithm, that is able to simulate only accumulated damage around voids or surface discontinuities). 

The stress gradient algorithm clarifies that damage starts from surface discontinuities or blood vessel (Figure 12) and not from lacunae, confirming from a computational point of view the hypothesis discussed in Section 3.

Despite the explained potentialities of linear micro-FE damage models, micro-damage predictions could be improved by implementing the bone non-linear behavior. The experimental tests focused on non-linear bone properties indicate ductile failure modes [112], in which damage is combined with the plasticity component of collagen fibers [113]. Hammond et al. [114] perform non-linear FE models (in this case, material non-linearity is considered) on trabecular samples from the distal femur of a human cadaver. They consider two micro-crack formation criteria, one for isotropic tissue models and one for anisotropic tissue models. For the isotropic tissue models, micro-cracks initiate if the ratio between maximum tensile principal stress and micro-crack initiation strength is equal to one. For anisotropic tissue models, micro-damage initiates if the maximum ratio between the normal of shear tractions with respect to the micro-crack initiation strength is equal to one. This non-linear micro-FE model is able to demonstrate that the anisotropy of bone tissue significantly contributes to bone fracture resistance [115]. Stipsitz et al. [116] try to implement an efficient micro-FE solver for large-scale non-linear simulations. However, the non-linearity of the system makes the results highly dependent on small structural deviations introduced by coarsening the structure. Additionally, one of the main drawbacks of the use of non-linear models is the increased computational cost, almost 10 times higher than linear simulations [87].

An increased degree of complexity can be found when analyzing nano-scale damage models that require further understanding of the variations in chemical pathways due to aging or disease. The first attempts in nano-scale bone fracture modeling were performed by Dubey et al. [117]: in their study, bone tissue fracture properties are based on the atomistic strength analyses of type I collagen in combination with hydroxyapatite interfacial arrangement, using molecular dynamics. Additional nano-scale studies [118] reveal that a decrease in the hydroxyapatite crystal size may change the mechanical behavior of the whole bone and the failure mode: from brittle-like crack-driven failure in larger crystals to a more widespread failure mode in the smaller ones. However, heterogeneities at the nano-scale remain difficult to model in a constitutive law and their role and effects are still unknown [119]. The impossibility to perform direct experimental tests on bone samples at the nano-scale is also a reason why those models are less considered for the purpose of this review.

To summarize, the increased image resolution and the optimization of parallel computational architecture for the solving of FE damage models seem promising tools for the improvement of the clinical understanding of fracture risk prediction. It is necessary to point out that some attempts in performing multiscale analyses have been performed by implementing bottom-up multiscale approaches, starting from the nano-scale up to the meso-scale [120] and analogously by Kwon et al. [121], who apply the multiscale analysis with variable geometrical parameters in order to determine its effect on the bone properties.

Before translating in silico damage models to clinical practice, the essential step is the experimental validation of the mentioned models that are still in their infancy, especially at smaller scales.

## 5. Validation Approaches to Multiscale Computational Damage Models

As regards the experimental validation of the mentioned multiscale FE models, preliminary studies are performed mainly following two paths: in vivo imaging [122] and dynamic image-guided failure assessment (dynamic IGFA) approaches [123]. 

The former allows the non-destructive monitoring of bone at multiple time intervals in a living animal. This could give relevant hints related to the effects of mechanical loading [124] and aging [125]. Despite the major developments of in vivo imaging in human subjects, it is not feasible to validate computational damage models on humans, due to the repeated in vivo high-resolution scans required to evaluate crack propagation. Further criticalities are then related to the radiation dose and to movements during the image acquisition, which reduce the repeatability of the test. Radiation exposure is one of the main limitations in in vivo micro-CT imaging, because it has been demonstrated that radiation causes additional damage to the bone [126]. The effect is particularly visible in the trabecular network, where a more sparsely connected trabecular network is present in radiated zones with respect to the unscanned ones [126]. Recent studies have tried to identify a precise scanning protocol able to find the best compromise between image quality and nominal radiation dose. Oliviero et al. [117] identify that a scanning regime of five scans every two weeks at 256 mGy has a limited effect on both mechanical and morphological properties of the considered region.

An alternative method, that overcomes the controversies of in vivo scans while preserving the possibility to validate computational damage models, is the dynamic IGFA approach. For this purpose, Müller’s group has developed an automated micro-compression device [127] for dynamic IGFA. It has been demonstrated that a micro-compression device for dynamic IGFA in combination with SR CT is a useful technique for dynamic high-resolution 3D assessment of bone damage propagation [123]. However, these studies are restricted to animal models. Performing dynamic IGFA experiments on human bone samples will better explain the interaction between the micro-architecture and fracture initiation and advancement. 

So, despite several studies on the characterization of micro-scale damage patterns, an effective validation of computational fracture models on human subjects is still lacking. This could lead to an improvement of the clinical fracture risk prediction, by obtaining micro-scale indices of bone fragility.

## 6. Conclusions

Assessing a patient’s risk of age-related fracture is a multiscale problem. Since age-related bone fractures represent both a health and an economic burden, early diagnosis should be fostered. Three main steps are key for improving fracture prediction: adequate choice of multiscale imaging technique, implementation of multiscale damage models and validation of those models.

As regards imaging techniques, DXA is commonly used to evaluate BMD. However, BMD is a limited predictor of fracture risk and it should be combined with observations related to bone quality. As an example, bone micro-scale morphological features are assessed by means of higher-resolution imaging modalities, whose radiation dose is, however, incompatible with clinical tests. 

The improved accuracy of imaging techniques permits the implementation of more detailed computational damage models that have the advantage of functioning as virtual experiments on the considered sample. In particular, micro-FE linear models seem to show the optimal balance between degree of complexity and computational costs. They are able to show some preliminary relations between micro-crack propagation and micro-scale morphology. A huge potentiality of those models lies in the capability to simulate the changes of bone microarchitecture and material properties in response to aging and disease.

However, the translation of patient-specific computational models from the computational side to clinical practice requires experimental validation that is still in progress. In particular, great potentialities are shown by the IGFA technique that is able to capture bone damage at different time steps. The actual state of the technology suggests that FE models could be a potential substitute for invasive clinical practice for the prediction of age-related bone fracture.

## Figures and Tables

**Figure 1 materials-14-01240-f001:**
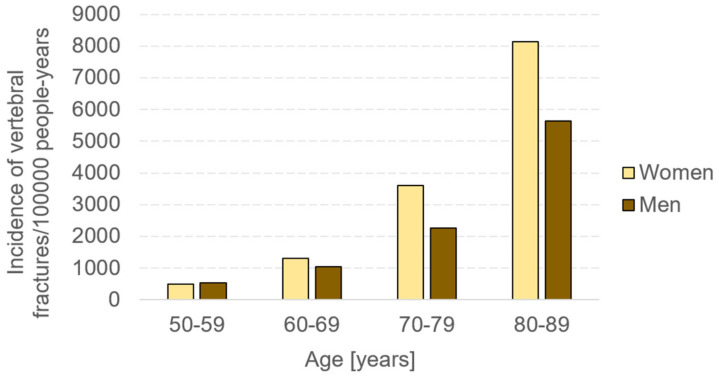
Incidence of vertebral fractures in Minnesota in years 2009–2010. The trend shows higher values for women >60 years old [5].

**Figure 2 materials-14-01240-f002:**
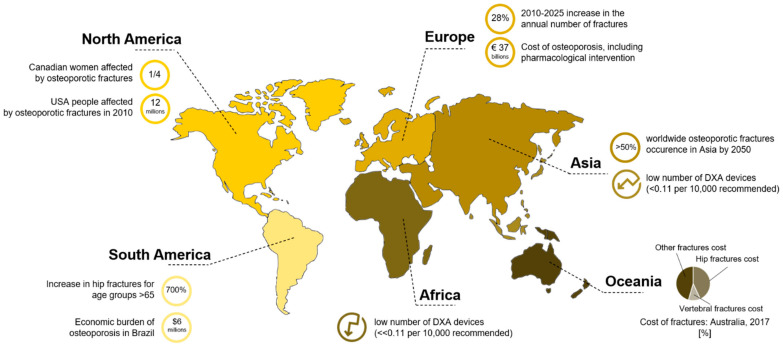
Infographic of the impact of osteoporotic fractures in the world. Australia: [9]. Rest of the world: [10]. DXA: dual X-ray absorptiometry.

**Figure 3 materials-14-01240-f003:**
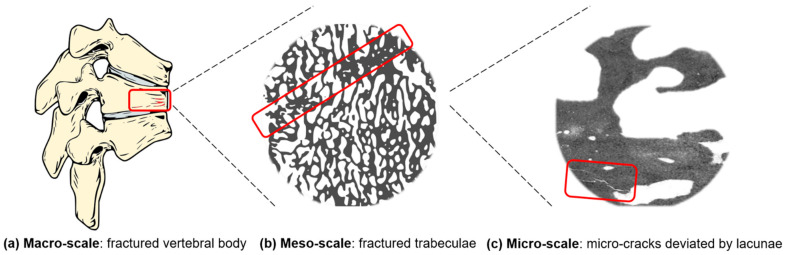
Schematic representation of the multiscale fracture mechanism (rectangular boxes). The macro-scale (**a**) is represented by a fractured vertebral body due to bone loss that amplifies the spine curvature. At the meso-scale (**b**), the fractured trabeculae are visible. At the micro-scale (**c**), the micro-crack starts from porosities and is deviated by the presence of the lacunae.

**Figure 4 materials-14-01240-f004:**
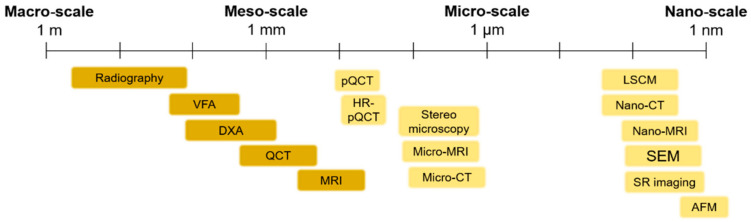
An overview of the main imaging techniques to assess bone damage at different scales. Macro- and meso-scale techniques are depicted in a darker color, while micro- and nano-scale imaging techniques are represented in a brighter color. VFA: Vertebral Fracture Assessment; QCT: Quantitative Computed Tomography; MRI: Magnetic Resonance Imaging; pQCT: peripheral Quantitative Computed Tomography; Micro-CT: Micro Computed Tomography; LSCM: Laser Scanning Confocal Microscopy; SEM: Scanning Electron Microscopy; SR imaging: Synchrotron Radiation imaging; AFM: Atomic Force Microscopy.

**Figure 5 materials-14-01240-f005:**
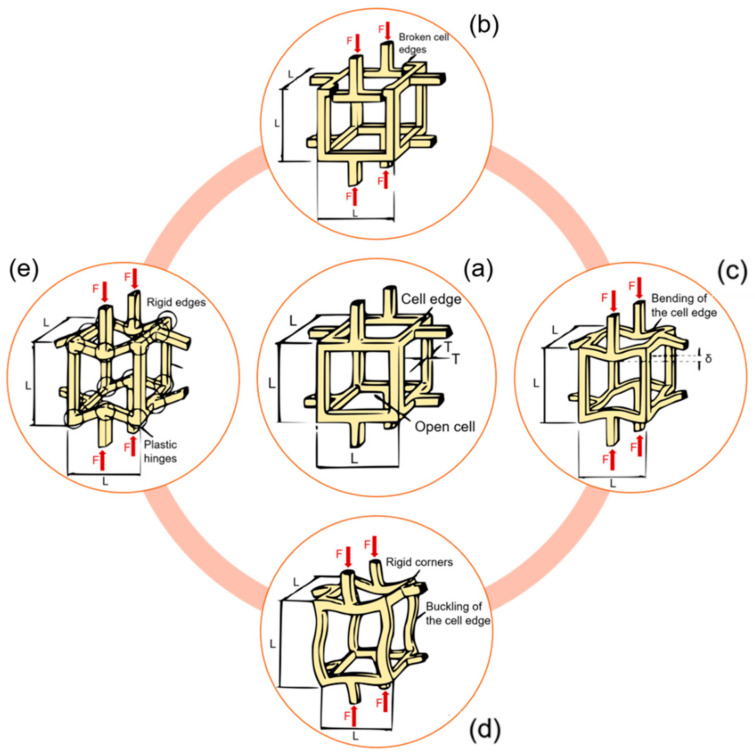
Different meso-scale failure modes of a trabecular network. (**a**) An unloaded trabecular cell. (**b**) Brittle crushing. (**c**) Bending of horizontal rods, compression of vertical rods. (**d**) Buckling of vertical rods, bending of horizontal rods. (**e**) Plastic yielding. Readapted from [68].

**Figure 6 materials-14-01240-f006:**
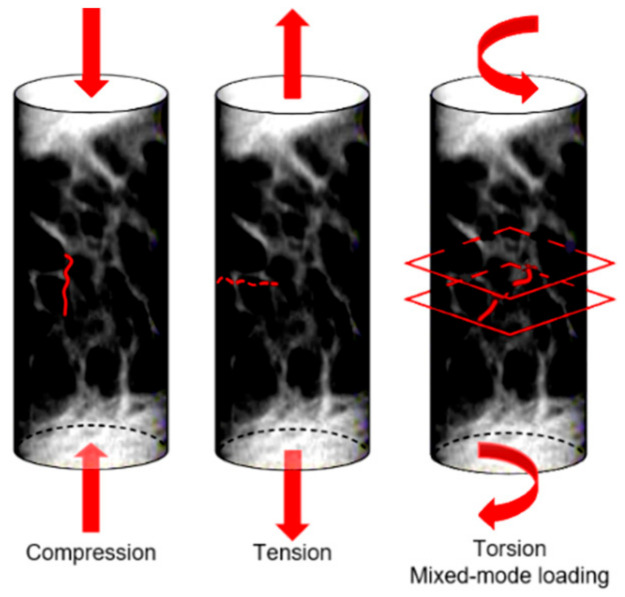
Orientation of micro-cracks in a bone sample subjected to different loading conditions.

**Figure 7 materials-14-01240-f007:**
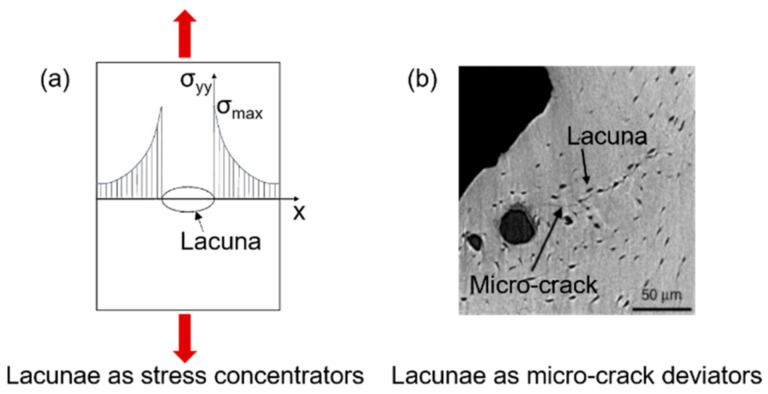
The dual effect of the lacunar system on microdamage: lacunae as sites for crack initiation (**a**) and lacunae as micro-crack deviators (**b**); (**b**) is adapted from [72].

**Figure 8 materials-14-01240-f008:**
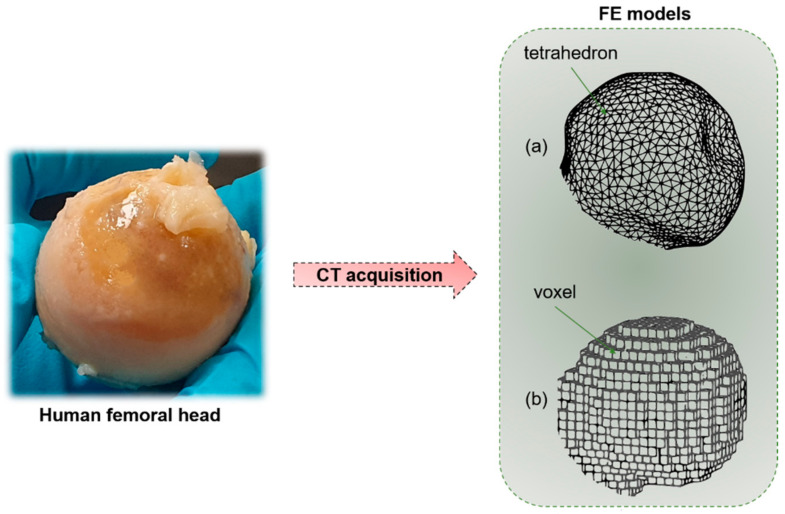
Starting from a human bone (e.g., femoral head), by means of CT acquisition, it is possible to obtain a macro-scale finite element (FE) model of the sample. (**a**) Geometry-based FE models with tetrahedral elements. (**b**) Voxel-based FE model with hexahedral elements.

**Figure 9 materials-14-01240-f009:**
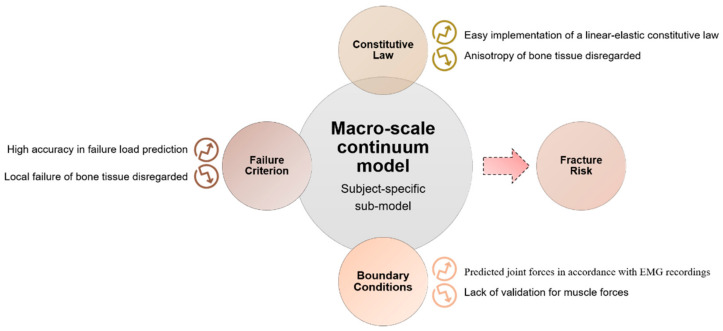
Scheme for the implementation of a whole-bone model. Results, potentialities and limitations are highlighted for each step. EMG: Electromyography.

**Figure 10 materials-14-01240-f010:**
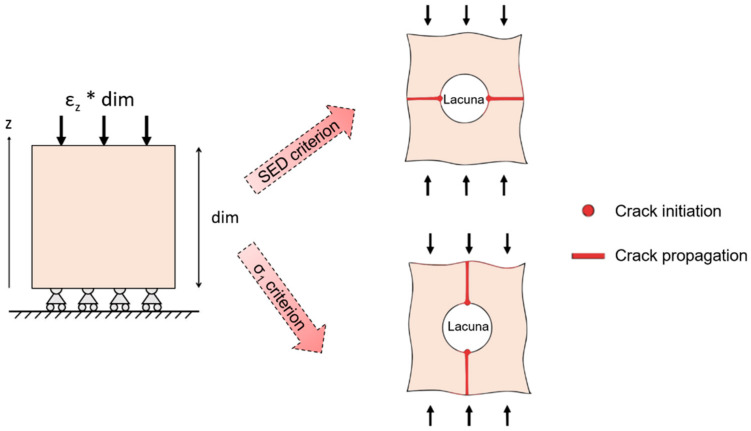
Schematic output of models of a bone sample with a lacuna subjected to compression. Crack initiation (dots) and crack propagation (lines) are highlighted, according to strain energy density (SED) damage criterion and to σ_1_ criterion.

**Figure 11 materials-14-01240-f011:**
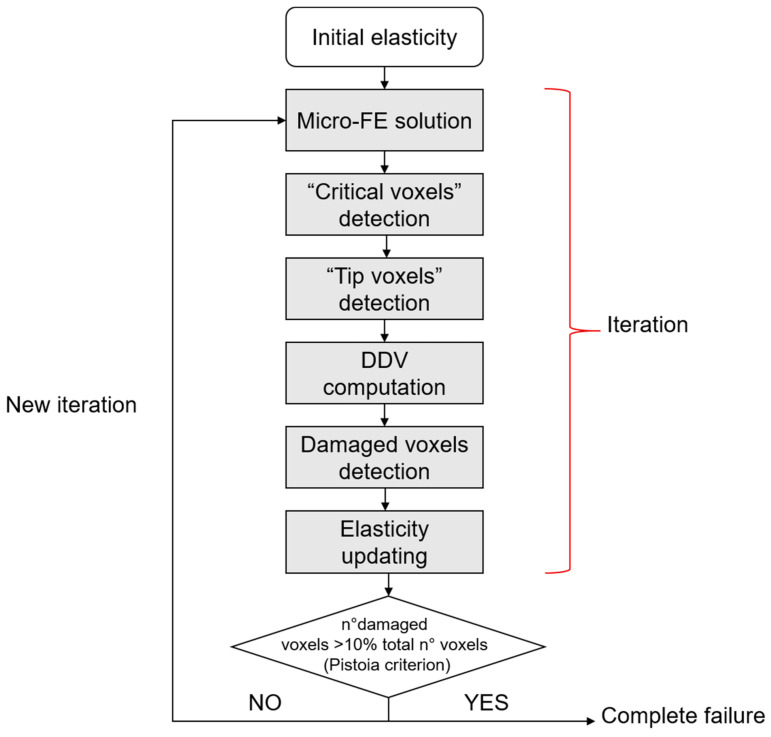
Stress gradient algorithm. “Critical voxels” are the voxels that exceed a defined stress threshold. “Tip voxels” are the center of the spherical sensation region of the osteocytes and are eligible sites for damage propagation. In the stress gradient algorithm, a key role is played by the deletion direction vector (DDV), that indicates a direction for crack propagation. The iteration stops when the total failure of the sample is reached, according to the Pistoia criterion [109].

**Figure 12 materials-14-01240-f012:**
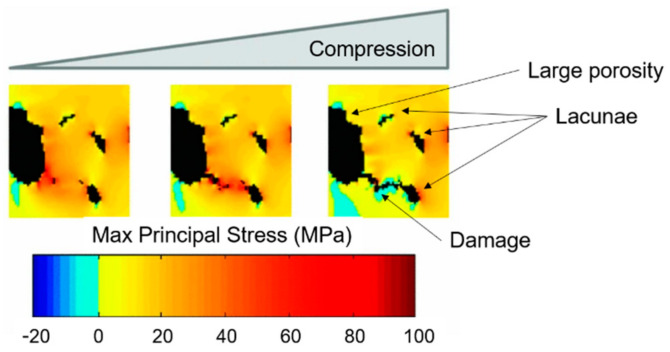
Stress gradient algorithm (on SR CT data from a cortical bone sample) results: damage starts from big porosities and its propagation follows the stress gradient. Micro-damage is deviated by the presence of lacunae. Readapted from [111].

**Table 1 materials-14-01240-t001:** Overview of the main macro- and meso-scale imaging techniques.

Macro- and Meso-Scale Imaging Technique	Brief Description of the Technique	Invasiveness	Outcomes	Spatial Resolution	2D or 3D	Advantages	Disadvantages
					In Vitro/In Vivo Application		
**Radiography**	Based on the interaction between a beam of photons (X-rays) directed from a source to a receptor. The atoms of the body prevent, in a percentage dependent on their atomic number, some photons from reaching the receptor, reproducing a “negative” image of the body	No Radiation dose: 40–50 times lower, if compared to computed tomography (CT) scans (e.g., radiographs of the abdomen → 0.25 mGy) [39]	Estimation of density variation (fracture risk prediction) by means of two indexes: Singh index [40] for proximal femur and cortical–medullary index [41] for hand radiographs	0.17 mm/pixel → The size of the monitor screens used in digital radiography is sufficient for 35 × 43 cm^2^ radiographs to be displayed at a resolution of 2048 × 2560 pixels [42]	2D	Clear identification of distal radius fractures [43]	Difficult detection of hip and spine fractures Insensitive to changes in Bone Mineral Density (BMD)until 20 to 40% of bone mass lost [43]
					In vivo		
**Dual-energy X-ray Absorptiometry (DXA)**	Involves the emission of two X-ray beams with different energy levels, that collide with the body of the patient. Once the absorption of the soft tissue has been subtracted, it is possible to determine the absorption of the beam by the bone and therefore the BMD	No Low radiation dose (0.001–0.003 mGy for L-spine, to 0.004 mGy for total body) [37]	Determination of areal BMD in g/cm^2^ Calculation of bone mineral content (BMC = BMD × area) Calculation of T-score and Z-score (negative for values under the average BMD), that are numerical indexes for the evaluation of osteoporosis.	1 pixel → ≃ 0.56 × 0.56 mm^2^. (for a Hologic system) [44]	2D	Ease of use of the equipment Standardization Short examination time [45]	No bone architecture detection (no difference between cortical and trabecular bone) Sampling errors Incorrect evaluation in obese patients [37]
					In vivo		
**Vertebral Fracture Assessment (VFA)**	Special DXA analysis that permits the detection of spinal fractures from a lateral image of the spine	No Lower radiation exposure with respect to spine radiography [46]	Spinal fracture detection [47]	Low spatial resolution	2D	Possibility to add a VFA scan after areal BMD assessment High sensitivity High specificity [48]	Low spatial resolution
					In vivo		
**Quantitative Computed Tomography (QCT)**	X-ray-based technique that measures BMD. It produces cross-sectional images of X-ray absorption coefficient (measured in Hounsfield units) calibrated to water. It is used to evaluate fracture risk primarily at the lumbar spine and at the hip [49]	Medium–high invasiveness Medium–high radiation dose (0.2–0.4 mGy for a spine exam) [50]	True measurement of BMD assessment (areal BMD does not predict if an individual patient will eventually fracture)	100× higher resolution with respect to conventional radiologic imaging [51]	3D (multiple slices are obtained and then reconstructed)	Fracture risk prediction in patients with scoliosis, obesity, etc. without having artificially high BMD values, as in DXA [52] High reproducibility Assessment of cortical and trabecular bone Good accuracy and precision [37]	Relevant radiation dose Low accessibility High cost [53]
					In vivo		
**Magnetic Resonance Imaging (MRI)**	MRIs employ a magnetic field that forces protons in the body to align with that field. When a radiofrequency current is pulsed through the patient, the protons are strained against the pull of the magnetic field. When the radiofrequency field is turned off, the MRI sensors detect the released energy as the protons realign with the magnetic field. The time it takes for the protons to realign, as well as the amount of energy released, changes depending on the environment and the chemical nature of the molecules	No MRI does not use ionizing radiation	Bone fracture detection Parameters: T2* [54] (effective transverse relaxation time) → a function of the density and orientation of the trabeculae [55] R2* → rate constant of the free induction signal (lower with respect to the control in osteoporotic women’s bone marrow [56])	MRI scanners used for medical purposes could reach typical resolutions of around 1.5 × 1.5 × 4 mm^3^ [57]	3D	Useful in age-related fracture detection (marrow fat increases with age and in osteoporosis, allowing better contrast with the trabecular bone) Investigation of cortical water content [43]	Presence of a magnetic field (risk for patients with pacemakers and all implants containing iron) Noise up to 120 dB Use of contrast agents Claustrophobia side effect
					In vivo		

**Table 2 materials-14-01240-t002:** Overview of the main micro- and nano-scale imaging technique.

Micro- and Nano-Scale Imaging Technique	Brief Description of the Technique	Invasiveness	Outcomes	Spatial Resolution	2D or 3D	Advantages	Disadvantages
					In Vitro/In Vivo Application		
**Stereomicroscopy Based on Histological Sections**	Histology from the bone tissue is obtained and then the sample is properly treated (fixation, dehydration and clearing, embedding, sectioning, staining and mounting). The histological section is then observed by means of an optical microscope	Yes	Traditional technique for the visualization of bone microarchitecture	~1.6 µm [58]	2D	Bone remodeling assessment [59]	Destructive and invasive technique Limitations related to the bidimensional output images: the three-dimensional features are lost. High-resolution images (at least 1.4 µm or better) are required to identify and measure individual resorption cavities in the process of bone remodeling [59]
					In vitro		
**Micro-Computed Tomography (Micro-CT) and Nano-Computed Tomography (Nano-CT)**	Micro- and nano-CT scans use radiographs to generate cross-sections of bone, that are generally processed (image reconstruction) to generate a virtual 3D model without destroying the original bone sample	No Generally, the samples are obtained from surgical wastes that derive from prosthetic treatment	Microarchitectural 3D data for both the cortical and the trabecular sections (tissue volume, bone volume, bone surface, bone volume fraction, bone surface to tissue volume, trabecular/cortical thickness, degree of anisotropy, cortical porosity, etc.) [37]. Local and global parameters related to the lacunar network are obtained [36]	1.2 µm (micro-CT) ~50–150 nm (nano-CT)	3D	Large number of obtainable outputs (morphological parameters at different scales) Detailed finite element 3D models could be implemented by using micro-CT images	Static evaluation of micro-scale features Not suitable for in vivo human evaluation due to the high radiation dose No detection of the canalicular network (insufficient resolution for the micro-CT scans) Nano-CT
					In vitro		
**Peripheral QCT (pQCT) and High-Resolution pQCT (HR-pQCT)**	Dedicated CT scanners for the forearm (radius and ulna) and leg (tibia and fibula)	No Low radiation dose (≈0.003 mGy) [37]	Analysis of the trabecular and cortical sections (BMD, bone mineral content and bone geometrical parameter calculation). Acquisition of biomechanical parameters, such as the cross-sectional moment of inertia. Evaluation of the functional muscle–bone unit [60].	Isotropic voxel size of 82 μm with HR-pQCT	3D	High precision and accuracy Low radiation dose Applicable for the study of a large number of diseases, especially pediatric (useful in applications where trabecular and cortical sections are affected in a different way)	Evaluation restricted to the appendicular bone Only transversal data are available for fracture risk prediction Low spatial resolution
					In vivo		
**Synchrotron Radiation Imaging (SR)**	A high-intensity white beam travels around a fixed closed loop. It permits a high level of detail in bone visualization (ultra-structural porosity detection)	No Generally, the samples are obtained from surgical wastes that derive from prosthetic treatment	Morphological analysis of ultra-structural porosities	Voxel size of 0.9 μm for the white beam [61]	3D	Visualization of the lacunar and canalicular network Phase contrast permits the clear detection of micro-cracks	Reduced field of view
					In vitro		
**Micro-MRI and nano-MRI**	The technique generates images by exploiting the nuclear magnetic behavior of different atoms in a sample tissue placed in a magnetic field	No	Structural parameters, such as trabecular bone thickness and mean bone volume fraction, associated with bone biomechanical properties and fracture resistance	Spatial resolution up to 25 µm (micro-MRI) and ~10 nm for the nano-MRI	3D	Non-destructive technique Good special resolution Good contrast resolution [62]	Long acquisition times High costs [62]
					In vivo		
**Laser Scanning Confocal Microscopy (LSCM)**	LSCM employs lasers at proper wavelengths to excite fluorochromes that are used to stain bone sections	Yes	Correlation between micro-crack parameters and bone matrix toughness Comparison among damage morphologies [13]	180 nm laterally and 500 nm axially [63]	2D/3D images of consecutive planes can be reconstructed into a 3D image in vitro.	Evaluation of bone microdamage	Axial resolution in depth impaired by spherical aberration [63] High costs
**Scanning Electon Microscopy (SEM)**	SEM produces images of the bone sample by scanning the surface with a focused beam of electrons	Yes	Quantitative analysis of fracture surfaces Visualization of microdamage morphology, fiber bridging and interlamellar separation [13]	~1 nm	3D In vitro	Significant information related to sub-micro-scale damage	Destructive technique (sample surfaces should be conductive → bone needs to be coated with conductive materials)
**Atomic Force Microscopy (AFM)**	The deflections of a cantilever on the surface of the bone sample are transduced into electrical signals	Yes	Topographical parameters of fractured bone surfaces (mineral particle sizes) Identification of sacrificial bonding	Vertical resolution → up to 0.1 nm Lateral resolution → ~30 nm	3D In vitro	Versatile imaging technique for the visualization of fracture surfaces High accuracy Non-destructive technique [64]	Small dimensions of the single scan image size (150 × 150 µm, compared with mm for SEM) Slow scan time [64]

## Data Availability

No new data were created or analyzed in this study.

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
