# Peer review of "A Review on Multiscale Bone Damage: From the Clinical to the Research Perspective"

_materials, 2021, doi:10.3390/ma14051240_

Round 1
Reviewer 1 Report
I recommend to accept the paper after minor revisions.
I would recommend to extend the last paragraph of chapter 2 and describe a possible future usage of imaging techniques in bone damage and to more discuss possible usage of in vivo and vitro imaging techniques.
It would be more straightforward to use the same units within the Table 1 in description of radiation dose.
Abbreviation IGFA is not described in the paper.
There is an incorrect page numbering in article (Table 1).
There is a inconsistent format of references.
Author Response
We would like to thank the reviewer for the detailed comments and for the positive feedback on our work.
The proposed suggestions have been implemented as follows:
- The last paragraph of chapter 2 has been extended with two main aims:
- to provide a general overview of all the details presented in Tables 1 and 2, summarizing the most relevant outcomes of the comparison between in vivo and in vitro imaging of bone fractures
- to provide future insights on the use of these imaging techniques: in particular, the necessity to better comprehend damage initiation and progression at smaller scales has been highlighted, with the aim to support early diagnosis of bone pathologies. The importance of in vitro imaging for the implementation of reliable and validated numerical models is also underlined.
- All the units of measurements used in Table 1 for the radiation dose have been converted to mGy
- The description of IGFA’s abbreviation has been added to the text.
- The page numbering has been updated
- The references have been formatted according to the journal’s style
Reviewer 2 Report
Dear Authors,
I carefully reviewed the manuscript entitled: A Review on Multiscale Bone Damage: from the Clinical to the Research Perspective. The topic you afford in this work which consider to application multiscale imaging techniques for implement reliable and validated numerical tool to prediction of bone fracture is very interesting.
Before the publication I would like to know few additional information:
- The part of the manuscript devoted to the comparison of multiscale imaging techniques used to assess bone fractures lacks a separate summary of data contained in Table 1 and Table 2. This would make it easier to receive the compiled data if in the subsection on Macro-and meso-scale imaging there would be conclusions about the quality of data collected by a given technique, their relevance to the models being developed. A similar summary is proposed to prepare for the subsection Micro-and nanoscale imaging. The current summary of tables 1 and 2 is too condensed.
- In the part of the paper concerning the physical damage of bones in different scales: the role of lacunae in the propagation/braking of micro-cracks also needs to be clarified. Material defects – pores are not always stress concentrators, they can also be inhibitors if the applied stress is less than the deformation of the movement limit material (which is additionally filled with fluid and osteocytes). It is therefore incomprehensible to state that; the lacunar system should contribute to strength decreasing all the more so as the Authors continue to write that lacunae are not the starting point for the micro-cracks that ultimately lead to bone failure.
- Are there any literature data that would show the use of the presented imaging methods in one work and would go deeper and deeper into the bone structure by analyzing its behavior in all the scales mentioned? The summarized authors show the use of data from one imaging technique and attempts at simulation on their basis. Does this mean that modelling of bone behaviour under load is only possible within the data collected during a single study, e.g. uCT? There is no such clear summary confirming or contradicting the proposed approach of large-scale analysis of bone behaviour under load can be used in modelling its behaviour during slow destruction on many levels.
It is also necessary same minor correction should be do in the manuscript:
- Tables 1, 2 with a smaller interline (such an extended text is difficult to read and takes up a lot of space, you cannot see a row with column headers)
- Figure 6 is required; unfortunately, the color and size used to show the damage are illegible especially in torsion mixed mode loading.
- Regarding references, please follow the same format according to the journal requirement (ie. 3,4,10,24,27,30,23,35,37,41,54,68,69,71,82). Some refs have not bold date, abbreviations and some have full form, maintain consistency

Reviewer 3 Report
Dear Authors,
Your article is well written and its publication brings you lots of quotations. Especially if other researchers are looking for experimental and comparative data for their research with the range of mechanical properties of osteoporotic bones or implantological treatment in orthopaedics. Introduction, the literature review is very good and there is no need to improve it. Only my comments concern the journal to which you want to publish this - MDPI Materials deals with more engineering issues, in the field of materials chemistry, solid body physics or material processes and research methods. If you want to make this article successful, it is not suitable here and your article will not be recognized. A better idea is to send this article to :
MDPI Medical Imaging ( secior of Diagnostics, ISSN 2075-4418)
https://www.mdpi.com/journal/diagnostics/sections/medical_imaging
Your article is about medical imaging so it does not fit here. For this reason, I do not recommend this article for evaluation and approval because it is not the subject of MDPI Materials. Try MDPI Medical Imaging.
I wish you good luck, sincerely
Reviewer
Author Response
We truly appreciate the reviewer’s feedback regarding the style and topics’ organization of our paper. However, we would like to underline that the aim of this extensive review is not just to analyze several medical imaging techniques, but to deeply understand and compare the current state of the art about the three main steps that should be followed for improving bone damage mechanisms’ comprehension and consequently fracture prediction. These steps, as mentioned in section 6 , are: 1) a suitable choice of multiscale imaging techniques, 2) the implementation of multiscale damage models and 3) their ultimate validation. In this perspective, imaging techniques at different length scales are essential tools for the study of bone’s hierarchical architecture from both the clinical and the research viewpoints and for the implementation of reliable numerical models, but they are not the only aspects to be considered for the comprehension of bone fracture processes. Our idea is to present a critical overview of the consolidated knowledge related to macro and meso-scale bone damage and to review and discuss the contribution of micro and nano-scale features (i.e., lacunae) that still is on debate in the current research landscape.
Reviewer 4 Report
this is just a narrative review and i don think is gonna add much to literature
Author Response
We would like to thank the reviewer for the provided comments.
The current review aims at providing critical insights to the investigation of bone damage at the multiscale, that is a crucial aspect to understand the mechanisms of age-related bone fractures. The authors address several aspects that are considered relevant for fracture prediction, a current, underexplored and debated topic in the clinical landscape, as demonstrated in section 1 (line 98 and following).
Firstly, a detailed comparison between in vitro and in vivo imaging techniques at the multiscale is presented. The invasivity, outcomes, resolution, main advantages and disadvantages are quantitatively highlighted: this will help the choice of suitable imaging techniques for the implementation of reliable numerical tools for fracture prediction.
Secondly, an extensive analysis of bone damage physical principles is presented: this aims at highlighting the current lacks in the research landscape, that could be overcome with the aid of numerical models. Thirdly, several works related to multiscale damage models have been categorized depending on the hierarchical scale they focus on, underlining their potentialities and limitations in shedding some light on damage initiation and propagation sites. Finally, validation approaches of the proposed models are critically compared, suggesting that computational damage models, in combination with suitable imaging techniques could be potential substitutes of invasive clinical practice for the prediction of age-related bone fractures.
Reviewer 5 Report
The paper presents a review on multiscale bone damage and combines perspectives from research and for clinical applications. Overall, the review is thorough and well done with an emphasis on imaging, mechanics, and modeling. Comments are as follows:
Tables 1 and 2 are very informative but also very complicated and long, the authors should consider ways to provide this data in a more organized and/or concise manner.
Figure 5 ordering is confusing in the ordering of the panels, it is recommended to revise with the panels moving from b, c, d, and e in a clockwise manner. Currently b and c are opposite to one another that is unexpected.
P3 line 89, why is tension considered the worst case loading scenario rather than compression, bending, torsion or other cases?
For Figure 8, could the authors please provide more insights for advantages/disadvantages of voxel or tetrahedron models, for instance when is it appropriate to use the voxel model?
P17 line 350, could the author’s please explain how non-linear behavior differs from linear behavior with some further examples of the type of non-linear relationships bone may follow?
P20 line 450 what is the difference in a linear and non-linear FE model?
Round 2
Reviewer 3 Report
Dear Authors,
Thank you for your factual reply. I 'm not able to dispute what you have written in your review because it is all valuable information and factual. I only meant that it does not quite fit into the profile of the journal to which you are trying to place it. However, if the editor-in-chief has no objections to the publication of this article, I am also in favor.
Thank you for your understanding and for your kind reply. In the present situation, I recommend the article for acceptance.
Bsest Wishes
Reviewer